# Museum Tour Guide Performance: A Visitor Perspective

**Željko Anđelković [1]**, **Sanja Kovačić [2,3]**, **Marija Bratić [4]**, **Miroslav D. Vujičić [2,\*]**, **Uglješa Stankov [2]**, **Vanja Pavluković [2]**, **Aleksandra Dragin [2]**, **Tatjana Pivac [2]**, **Anđelija Ivkov Džigurski [2]**, **Ljubica Ivanović Bibić [2]**, **Zrinka Zadel [5]** and **Smiljana Đukičin Vučković [2]**

1. National Museum Niš, 18000 Niš, Serbia
2. Department of Geography, Tourism and Hotel Management, Faculty of Sciences, University of Novi Sad, 21000 Novi Sad, Serbia
3. Institute of Sports, Tourism and Service, South Ural State University, 76 Lenin Ave., 454080 Chelyabinsk, Russia
4. Department of Geography, Faculty of Science, University of Niš, 18000 Niš, Serbia
5. Faculty of Tourism and Hospitality Management, University of Rijeka, 51000 Rijeka, Croatia
* Correspondence: miroslav.vujicic@dgt.uns.ac.rs

**Abstract:** Tour guide performance has been a hotly debated academic topic, owing to the critical role they play in facilitating a variety of tourist experiences. Similarly, museum tour guides are the initial point of contact for tourists and have a substantial impact on the total visitor experience, but their performance is far less investigated. Visitors' satisfaction and behavior intentions are inextricably linked to museum guide performances in this case. The purpose of this research was to implement and test a well-established scale for assessing tour guide performance in the museum context. Moreover, the aim was also to single out and discuss museum guide types based on their performances. The research was conducted during May and June 2021 on a sample of 255 visitors from five museums in Serbia. The data were processed by SPSS, R and RStudio. The results indicate the existence of five museum guide types: Classic Professional, Agile Empath, Operational Erudite, Trustworthy Caretaker, and Passionate Socializer. The study explores sociodemographic differences in visitors' evaluations of tour guide performances and gives theoretical and managerial implication for each museum guide type.

**Keywords:** museum tour guides; tour guide performances; tour guide types; cultural heritage; Serbia

## 1. Introduction

Museum tours are an essential part of the museum visitor programmes [1], with museum tour guides being a pivotal part of them. Literature focusing on museum guides is quite scarce, with many research questions still waiting to be answered. According to Peng and Chen [2] a fast-growing popularity of research focusing on mobile guides and digital devices influenced the fact that research on museum tour guides is quite neglected. The majority of the existing studies on museum guides is in fields of tourism, emphasizing their role in creating overall tourist satisfaction and behavioral intention [3]. However, visitors' perspective, and their interaction with tour guides, was often neglected in the early papers [1,4,5]. The study of Peng and Chen [2] also argues that the museum tour guides are undervalued and calls for more research in this field, especially due to the importance of live tours for better on-site experience in a post COVID-19 period.

The sustainable museums of the 21st century are energy efficient, sustainably managed and raise public awareness of environmental issues [6] and in that sense raise public awareness of the great importance of cultural heritage and its conservation and interpretation. The big role in a better understanding of museums' roles in society and of the importance of cultural heritage have museum tour guides with their stories and their performances. There are many means for museums to express this social, progressive component of education: museums can support growth and development for all individuals—democracy and

human rights for all—in plenty of ways [7] and one can be educated about sustainability; museum tour guides together with curators are actually those who have the closest contact with visitors. This is why research on museum tour guides is important in the light of sustainability of museums.

The few studies focusing on museum guides cover several frequently studied topics—museum guides' expertise [8], the museum guide strategies [9], the nature of guided tours [1], or competences of tour guides [10–13], as well as their education and professionalization [1,14]. Due to the lack of literature on museum guides and their performance, the important theoretical background can be found in tour guide literature in tourism.

In the tourist industry, tour guides are the ones who are in direct contact with tourists and those who mostly influence their perception and experience of a certain destination [15]. Similarly, museum tour guides are first to interact with visitors and have a large impact on overall visitor experience. Tour (museum) guide interpretation is the base for heritage interpretation [16] and in that sense satisfaction with museum tour guides is tightly connected with their performances and skills. While tour guide performance and its relationships with tourist satisfaction and behavioral intentions has been largely studied in the tourism field [3,17–21], the performance of museum guides is still underexplored. Moreover, there are no papers measuring or applying tour guide performance scales in the museum context. Thus, taking into account the discussed literature gap, this paper tends to contribute to the research in the area of museum guide performance. The principal aim of the paper is to apply and test a tour guide performance scale developed by Huang et al. [15] in the specific context of museum guides. Moreover, the paper aims to single out the types of museum guides based on their performances. The paper is based on the belief that visitors' satisfaction and their behaviour intentions are tightly knit with museum guide performances. This is why the paper studies museum guide performances from visitors' perspectives. Finally, the study explores sociodemographic differences in visitors' evaluations of tour guide performances. Theoretical and managerial implications are discussed in the paper.

## 2. Literature Review

### 2.1. Research on Museum Guides

Despite the importance of tour guides for tourist activities, studies on tour guides, particularly within the museum sector, are quite rare. According to Best [1], this is why we lack understanding of the guided tours and museum guides themselves.

Tour guides are of a paramount importance for generating tourist satisfaction and revisiting intentions [3,15,17,22,23]. Cohen [24] and Huang et al. [15] also argue that tour guides, as frontline workers, are important for creating quality customer experience. Tour guides are also deemed to be storytellers creating memorable experiences for tourists [21,25–27]. However, the current study is based on the assumption that a tour guide is not the same as a museum guide and that visitors perceive and expect different performances from them. This is why we decided to test the tour guide performances scale in the museum context and see what the structure of the guide performances is in this sense.

The role museum guides have in guided tours is an important research area in this field. Early research on museum tours and tour guides is mostly directed towards explaining how to interpret artworks, while later, museum guides have gained much more importance associated with the different roles they have in creating tourism experience [28,29]. Tour guides are mostly considered as skillful storytellers [21,25–27]. Moreover, Best [1] emphasized the work of Cohen [24], which identified two important roles: pathfinder (guiding an audience around a site) and mentor (giving all the information that can be interesting for visitors on a site). Additionally, tour guides have the important role in answering different needs, interests, and wants of various tourists, being expected to be flexible with the tour schedule [11]. During a museum tour, guides take care of the proper movement of the group [11], ensuring safety of the artefacts [12]. Moreover, the tour guides are seen as heritage protectors, cultural mediators, sustainability promoters, site interpreters and behavior modifiers [15,30–32]. As summarized by Chilembwe & Mweiwa [33] and

Orabi & Fadel [34] the tour guide performance research generally encompasses three areas: Tour Management (Organizer-manager, Entertainer), Experience Management (Leader, Cultural broker, Educator, Information giver) and Resource Management (Motivator or Mentor, Interpreter or Mediator).

Another important research topic about museum guides is related to their competences [10–13]. As museums are very educative places, tour guides must possess different competences needed to ensure a quality tour. For example, guides should be able to make visitors be more focused on the object [10,35], to encourage debate about objects [10], and to help in analyzing historical relevance and background [36]. Moreover, guides should have a combination of competences, skills, knowledge, and behavior, in order to provide a quality tour [18]. For instance, understanding different cultures and traditions of the community is also an important skill if they want to entice visitors to learn more about a museum, heritage site or another tourist site [37,38]. Some studies also emphasize the importance of the right vocational training as well as the professional education of tour guides [1,14].

On balance, the majority of the museum guides studies are related to their competences and the roles they have in the overall museum experience, while there are no studies particularly focusing on museum guides' performances. This is why we find the theoretical background for studying this topic in the rich literature on tour guide performances.

### 2.2. Tour (Museum) Guides Performances

Apart from basic guide performance which delivers service quality, tour guides have a power to immerse visitors into experiences by some special performances [19]. According to Hansen & Mossberg [19], this means that the guide focuses on each individual and its experience, playing a role of a storyteller, social mediator and instructor.

Literature suggests that a tour guide's performance is of paramount importance for tour success and tourist satisfaction [17,20,39,40]. Moreover, tour guide performance, apart from tourist satisfaction, has positive effects on perceived credibility trust and perceived benevolence trust [22]. According to Alazaizeh et al. [21], tour guide performance also affects the relationship between tourist satisfaction and behavioral intention. A tour guide with high performances will create a positive word of mouth, also affecting tourists' willingness to revisit a destination [22]. The importance of tour guides and their performances are also emphasized by the study of Çetinkaya and Öter [37], who revealed that the lack of personality shown by the tour guides negatively affects tour satisfaction and their willingness to revisit a destination. The article of Alazaizeh et al. [21] also showed that the performance of a tour guide enhances visitor sustainable behaviour. Weiler and Walker [41] argued that the communication skills of tour guides enhanced the overall tour experience. Weiler and Yu [42] emphasized three main roles of tour guides: the mediation of access, understanding, and encounters, with mediation of understanding being the most important for tourist experience. Due to these critical roles of the tour guiding performance for service quality and tourist satisfaction, special intention should be paid to enhancing tour guide performances through education and training [43].

### 2.3. Measuring Tour (Museum) Guides Performances

When it comes to measuring tour guide performance, there are several papers dealing with this topic. Huang et al. [15] developed and tested a scale of tour guide performances and extracted the following factors: Professional Competence, Interpersonal Skills and Organization, Empathy, and Problem-Solving. Hurombo [44], in his doctoral thesis, developed a scale of tour guide performances based on the tour guide competences and derived four factors: Cultural intelligence, Tour management, Professionalism and Social skills. Sezgin and Duz [43] have developed and tested the tour guide performance (GuidePerf) scale for tourism in relation to various tour guiding diplomas. The scale has 18 items in three dimensions, namely: Personality and efficiency, Presentableness, and Proficiency. In this paper, the measuring of tour guides' performances has been done from a visitor perspective, evaluating three different guides with different diplomas. In addition, Le

Nguyen [45] showed that tour guide performance consists of five factors—appearance, professional competence skill, solving problems skill, organizational skill, and entertainment introduction skill. It is also important to mention Hansen and Mossberg [19], who have extracted two factors: Interpersonal servability and organization and Intrapersonal servability and professional competence. A review of the available literature on tour guide performances led us to the conclusion that the majority of studies extracted the professional skills of tour guides but also emphasized some personal skills such as cultural intelligence, social skills and personality. Moreover, a review of the relevant studies suggests that tour guide performance was assessed by tourist evaluation in the majority of papers. Although there is a lot of literature on overall tour guides performance, there is an evident gap in literature on specialized museum guides' performances. With this in mind, the paper tends to test and apply a tour guide performance scale by Huang et al. [15] and will evaluate visitor perspectives on tour guide performance in the specific museum context.

## 3. Methods

### 3.1. Study Sample

The sample was convenient, including all museum visitors who were willing to participate in the research and were older than 18 at the moment of conducting research. The study sample consists of a total of 255 visitors of museums—123 domestic (visitors from Serbia) and 132 foreigners. There is a higher number of female respondents in the sample (51.8%), while the majority of visitors are in the 18–30 age category. Most of the visitors are employed (63.1%) and had finished secondary school (27.1%) or acquired a BSc. diploma (36.9%) (Table 1).There are more domestic, Serbian visitors (51.8%) than foreign ones, while the majority of the foreign visitors came from Europe, as presented in Table 2. The study included both visitors who have visited the museum individually and those coming as a part of organized groups.

**Table 1.** Sociodemographic characteristics of respondents (*N* = 255).

| Gender | | Education | |
|---|---|---|---|
| Male | 48.2% | Elementary school | 1.2% |
| Female | 51.8% | Secondary school | 27.1% |
| **Age** | | College | 13.3% |
| 18–30 | 33.8% | Faculty BSc | 36.9% |
| 31–45 | 32.5% | MSc or PhD | 21.6% |
| 46–60 | 20.4% | | |
| 61+ | 13.3% | | |
| **Average Monthly Income** | | **Work Status** | |
| No income | 18.4% | Employed | 63.1% |
| Below average | 17.3% | Unemployed | 8.6% |
| Average | 18.4% | Student/pupil | 16.1% |
| Above average | 45.9% | Retired | 12.2% |

**Table 2.** Country of origin of foreign tourists.

| Country | % | Country | % |
|---|---|---|---|
| Australia | 0.4 | Hungary | 4.7 |
| Austria | 0.8 | Italy | 1.2 |
| Bosnia—Herzegovina | 1.2 | Montenegro | 0.4 |
| Bulgaria | 9.4 | North Macedonia | 0.4 |
| Canada | 1.2 | Poland | 0.4 |

**Table 2.** *Cont.*

| Country | % | Country | % |
|---|---|---|---|
| Croatia | 0.8 | Romania | 3.5 |
| Czech Republic | 0.4 | Slovenia | 3.9 |
| Germany | 3.1 | UK | 7.8 |
| Greece | 0.4 | USA | 8.2 |
| Total | | 48.2% | |

### 3.2. Instruments

The questionnaire is comprised from two parts. The first part referred to sociodemographic characteristics of respondents (gender, age, education, work status and income) and also included a question on frequency of travel. The second part of the questionnaire measured tour guide performance from a visitor perspective (33 items), which were derived from the previous work of Huang et al. [15]. A five-point Likert scale (1—I totally agree, 5—I totally disagree) was used to measure the answers. The original scale, developed by Huang et al. in 2007, contained 35 items measuring the tour guide performance as perceived by Shanghai tourists. The following scale: 1 = extremely poor and 7 = extremely good was adopted to assess respondents' ratings on the performance of tour guides. The total sample of the original study included 366 copies of a Chinese questionnaire and 342 copies of an English questionnaire. The study applied EFA to derive factors from the tour guide performance attributes. From the total of 35 items, only two items were not included in the study as they were deemed irrelevant for museum guides. Two questions were omitted in the current questionnaire: the first question related to the knowledge of the culture and history of the destination was merged into one question because the culture and history of the destination in museums are largely connected. The second question—the tourist guide is always available when needed by the tourists—was omitted for two reasons: the first is because this question cannot be related to the performances of tour guides in museum, and the second reason is that the essence of this question is already part of several other questions in the questionnaire.

### 3.3. Procedure and Data Analysis

Data was collected through face-to-face standard paper-and-pen questionnaire of tourists/visitors to the Museum of Yugoslavia and Museum of Nikola Tesla in Belgrade, National Museum of Nis, and Museum of Vojvodina and Gallery of Matica Srpska in Novi Sad (Serbia). The questionnaire was conducted during May and June 2021.The data collection was conducted by some of the authors and partly by museum tour guides and museum staff who were on site at the time. The authors have permission from all museums for data collection.

The questionnaire was distributed at the museums after the tour was completed. Each interview took approximately 10–12 min to complete. In order to avoid misinterpretation from the foreign tourists, the questionnaire in English was read for them and them interviewer filled in their answers. Domestic tourists filled in the questionnaire in Serbian language by themselves. When they consented to filling in the questionnaire, the interviewers explained that all the questions are related to the museum tour guides rather than their tour guides when they were members of the group. It is important to mention that, in the museums where the research was performed, there was no option of audio guide—so all visitors only had the option to hear information from the museum tour guide.

During this process, a total of 268 questionnaires were completed, but 13 had missing values and were excluded from the further analysis. Thus, 255 valid answers were collected. We used exploratory factor analysis to extract dimensions of the Tour Guide Performance Scale (TGPS) from a visitor perspective. Confirmatory factor analysis was used to test the measurement scale; the maximum likelihood method of structural equation modelling was used to test the tour guide performance structure [46,47]. Construction validity was

controlled by checking the load of each item on the construction variables and using the fit index to test the model fit. EFA analysis, as well as Independent T-test and One-way ANOVA, were performed in Statistical Package for Social Sciences Version 23-SPSS (IBM Corp.: Armonk, NY, USA), while CFA analysis was done in R and RStudio (lavaan, semPlot, psych and semTools packages).

## 4. Results

### 4.1. Exploratory Factor Analysis—Museum Tour Guide Performance Scale (mTGPS)

Exploratory factorial analysis (EFA) with Varimax rotation has been done to extract the factors of the tour guide performance (Table 2). A Kaiser–Meyer–Olkin measure yielded 0.916, and Bartlett's test of sphericity was significant: 4776.535 (df = 528, $p < 0.000$). Five factors were extracted with the total of 57.965% variance explained. Cronbach's $\alpha$ ranged from 0.756 to 0.894, showing high reliability of the instrument (Table 3). According to the factor loading scores for each item, five different tour guide performance factors were extracted (Classic Professional—CP, Agile Empath–AE, Operational Erudite—OE, Trustworthy Caretaker—TC, Passionate Socializer—PS).

**Table 3.** Descriptive statistics of the latent variables and comparison with scale developed by Huang et al. [15].

| Factors and Items | Factor Loadings | Eigen Value | Cronbach's $\alpha$ | Variance Explained (%) |
|---|---|---|---|---|
| **Classic Professional—CP** <br> Dimension in original scale: Professional Competence and Interpersonal skills | | 12.737 | 0.894 | 16.505 |
| CP1 tries his/her best to follow itinerary and daily schedules | 0.474 | | | |
| CP2 is polite | 0.625 | | | |
| CP3 grooming and appearance are neat and appropriate | 0.552 | | | |
| CP4 is able to generate rapport among tour group members | 0.622 | | | |
| CP5 is good at interpersonal communication | 0.687 | | | |
| CP6 performs well in commentary | 0.587 | | | |
| CP7 is punctual | 0.594 | | | |
| CP8 is honest and trustworthy | 0.555 | | | |
| CP9 shows a sense of responsibility | 0.536 | | | |
| CP10 can follow the code of ethics in the profession | 0.614 | | | |
| CP11 is willing to help | 0.531 | | | |
| **Agile Empath—AE** <br> Dimension in original scale: Empathy, Problem-solving and Interpersonal skills | | 2.081 | 0.879 | 15.676 |
| AE1 puts himself in the shoes of customers | 0.458 | | | |
| AE2 is able to meet customer's psychological needs | 0.598 | | | |
| AE3 is capable of solving problems and conflicts emerged from tour arrangements | 0.762 | | | |
| AE4 is capable of handling customer complaints properly | 0.730 | | | |
| AE5 is able to cope with unexpected urgent incidents | 0.776 | | | |
| AE6 shows sound judgment in most situations | 0.745 | | | |
| AE7 understand the culture of customers he/she is serving (she/he is familiar with the area the country of their origin thus understands some possible specific things due to it) | 0.562 | | | |
| **Operational Erudite—OE** <br> Dimension in original scale: Professional Competence, Interpersonal skills and Problem-solving skills | | 1.843 | 0.858 | 9.760 |
| OE1 has the knowledge of the destination's culture and history | 0.710 | | | |
| OE2 is able to organize tour-related activities | 0.710 | | | |
| OE3 is good at time management | 0.591 | | | |
| OE4 is proficient in the tour-guiding language | 0.586 | | | |
| OE5 has the knowledge of local people's lifestyle | 0.555 | | | |
| OE6 has the knowledge of tourist attractions | 0.480 | | | |

**Table 3.** *Cont.*

| Factors and Items | Factor Loadings | Eigen Value | Cronbach's α | Variance Explained (%) |
|---|---|---|---|---|
| **Trustworthy Caretaker—TC** Dimension in original scale: Interpersonal skills and Professional Competence | | 1.301 | 0.762 | 8.535 |
| TC1 takes care of customer's needs(whatever is related to their stay at the destination) | 0.504 | | | |
| TC2 keeps reminding tourists of safety issues | 0.683 | | | |
| TC3 introduces reliable shops for customers | 0.739 | | | |
| TC4 has good health | 0.694 | | | |
| TC5 is able to cooperate with other service staff | 0.713 | | | |
| **Passionate Socializer—PS** Dimension in original scale: Professional competence and Empathy | | 1.167 | 0.756 | 7.398 |
| PS1 is friendly | 0.693 | | | |
| PS2 shows good sense of humour | 0.666 | | | |
| PS3 has good personality | 0.442 | | | |
| PS4 shows passion for his/her work | 0.404 | | | |

So, when we put the original scale to the different context—museum, the items constituted different factors and represented different types of tour guides based on their performances. Below each museum guide type in Table 3, the authors have stated the original dimensions the items were part of (in the original model). This finding suggests that all items are important in the museum context as well, but they have different structures—they constitute different performance types of tour guides.

*4.2. Measurement Model Validity for Museum Tour Guide Performance Scale (mTGPS)–Confirmatory Factorial Analysis*

Confirmatory factorial analysis (CFA) was used to check validity and reliability of the measurement model. Initial model fit indices showed good results and good fit indices, except TLI, which was just below threshold (CFI = 0.952 (>0.95), TLI = 0.948 (>0.95), RMSEA = 0.0057 (<0.08), SRMR = 0.075 (<0.08), df = 485, $p < 0.000$), indicating potential problems with the model fit. Therefore, modifications were consulted and showed that one item with high residual should be excluded (TC1), thus defining the model with satisfactory fit (CFI = 0.960, TLI = 0.956, RMSEA = 0.054, SRMR = 0.072, df = 454, $p < 0.000$). The final scale for visitors' perceptions of tour guide performance factors included five latent factors, with 32 items in total.

Scale reliability was assessed through Composite reliability (CR) and Average variance extracted (AVE) indices (Table 4). Average variance extracted (AVE) was also used to examine the convergent validity of each dimension [48]. A convergent validity is achieved when the AVE score is higher than 0.50 and (CR) is higher than 0.60 within each dimension [48,49]. Results showed that all dimensions had AVE higher than 0.50 and CR higher than 0.60 (Table 4), which indicates good convergent validity.

**Table 4.** Reliability of the instruments mTGPS.

| Constructs | AVE | CR |
|---|---|---|
| Classic Professional—CP | 0.57 | 0.94 |
| Agile Empath—AE | 0.66 | 0.93 |
| Operational Erudite—OE | 0.68 | 0.92 |
| Trustworthy Caretaker—TC | 0.57 | 0.84 |
| Passionate Socializer—PS | 0.59 | 0.85 |

Fornell and Larcker [48] noted that the discriminant validity is achieved when the square root of each AVE is greater than the correlation coefficients estimate, which is

the case in this study (Table 5). Thus, results confirm that all dimensions have sufficient discriminant validity [48,50].

**Table 5.** Discriminant validity mTGPS.

|     | CM    | AE    | OE    | TC   | PS   |
|-----|-------|-------|-------|------|------|
| CP  | *0.76* |       |       |      |      |
| AE  | 0.708 | *0.81* |       |      |      |
| OE  | 0.706 | 0.692 | *0.82* |      |      |
| TC  | 0.458 | 0.413 | 0.376 | *0.75* |      |
| PS  | 0.723 | 0.65  | 0.644 | 0.358 | *0.77* |

AVE value is presented in Italics.

*4.3. Sociodemographic Differences on the Museum Tour Guide Scale*

The discriminativeness of the scale was tested regarding participant gender, age, education, work status, and frequency of travel in relation to mTGPS scale. Independent T-test showed certain differences among gender. Females value more Classic Professional (CP) tour guides then males (t = −0.995, *p* = 0.000), and females slightly valued more Agile Empath (AE) tour guides then males (t = −0.830, *p* = 0.003). Using One-way ANOVA and post hoc LSD Test, further differences were found in regard to age and work status, while no statistical significance was found in relation to education, income and frequency of travel. For example, respondents in the age group (65+) most value the Trustworthy Caretaker (TC), followed by the respondents in the age by the age group (18–29) and lowest value by the age group (30–39) and (40–49) age group (F = 2.348, *p* = 0.042). In regard to work status, retired people value more the Trustworthy Caretaker (TC) factor then employed respondents (F = 4.260, *p* = 0.015).

**5. Discussion**

This paper aimed to apply and test a tour guide performance scale within the specific museum context, which was followed by exploring sociodemographic differences in visitors' evaluations of tour guide performances. The study applied an already validated scale developed by Huang et al. [15] that derived four dimensions, such as Professional Competence, Interpersonal Skills, Empathy and Problem-solving. Those dimensions were named and described as core competences and skills for tour guides in general.

The applied scale is a blend of items representing not only competences and skills, but also personality traits and overall impressions. Based on this, our study provides a different approach in explaining and naming extracted dimensions, looking at them as performance modes/types. Thus, the current study suggests five dimensions that are perceived and described as performance types of museum tour guides. The first performance type is named *Classic Professional (CP)* and includes a mixture of items that coincide professional competence and interpersonal skills from the original model of Huang et al., [15]. Our first type is described as person that is polite, follows an itinerary, has good communicational skills, is a responsible person and especially follows the code of ethics and professional guidelines, and can be perceived as classical role model.

The second performance type is entitled *Agile Empath (AE)* and includes a combination of Empathy, Problem-solving and Interpersonal skills from the original model from Huang et al. [15]. This type is described as a person that meets customer needs, puts himself in the shoes of customers, and understands the culture of customers, while also being good at solving problems and conflicts, handling customer complaints, knowing how to cope with unexpected incidents, and showing sound judgments in most situations. He is perceived as an emphatical problem solver.

The third type is called *Operational Erudite (OE)* and includes mixture of Professional competencies, Interpersonal and Problem-solving skills from the original model presented by Huang et al. [15]. The third type of museum tour guide is described as a knowledgeable

person that is good at organizing time and activities and is proficient in tour-guiding language. They have both organizational and knowledge skills.

The fourth type is titled *Trustworthy Caretaker (TC),* which is a blend of Interpersonal skills and Professional competence from the original model introduced by Huang et al. [15]. This type of museum tour guide has a set of different values such as taking care of customers' needs, reminding tourists of safety issues, and introducing reliable information, and is well connected with other service staff.

The fifth type is termed as *Passionate Socializer (PS),* which includes a combination of Professional competence and empathy from the original scale introduced by Huang et al. [15]. The fifth type of museum tour guide has a set of different characteristics such as friendly appearance and good sense of humor and shows passion for his work.

When museum tour guides are considered, we can detect significant distinctions among tour guides in general as five totally different types were defined within the study. These five types represent a combination of dimensions extracted in the previous work of Huang et al. [15]. Newly defined types of museum tour guides are regarded as a blend of previously defined tour guide dimensions from the original work of Huang et al. [15], which can be acknowledged as museums are discovered to be specific tourism destinations/contexts, requiring different core values for tour guides.

Additionally, both domestic and foreign visitors filled in the questionnaire in several museums in Serbia, and through statistical analysis, differences in visitor perception, looked at from the perspective of sociodemographic characteristics, are found in relation to five defined types of museum tour guides. Differences within age category and work status were found to have statistical significance for the Trustworthy Caretaker (TC) type. It is found that older (65+) and younger people (18–29) value Trustworthy Caretaker more than other age categories, which are indeed groups that may require more attention, information and deeper connection with the tour guide. The study also showed that retired people value TC more than employed people. This coincides with the finding that older people are those who value TC more than others, but it can be also explained by the fact that retired people have more time for museum exploration and require more attention from the tour guides. Such a finding can have a practical implication in museums, which is discussed further in the paper.

## 6. Conclusions and Further Research

The principal purpose of the study was to apply the tour guide performance scale developed by Huang et al. [15] in the museum context from the visitor perspective and single out the profiles of museum guides based on their performances. Moreover, the aim was to test socio-demographic differences in relation to developed museum tour guide types. The study contributes to the scarce knowledge of museum tour guides, who have a crucial role in raising public awareness of the importance of cultural heritage, contributing in that sense to the sustainability of museums as public institutions responsible for the presentation, interpretation and preservation of cultural heritage. The main theoretical contribution of the study is the fact that it addresses the research gap in the field of museum guide research, as this is the first study to apply a tour guide scale in the museum context. This is also the first study to describe the performance types of tour guides and their specificities, as the previous studies extracted dimensions representing skills, competences or capabilities. The current study argues that the blend of such characteristics can be represented by tour guide performance types. Practical implications can be found in audience development strategies, as audience development strategies often include the adjustment of cultural content to different age categories of visitors [51]. Different age categories can be overlapped and matched with different museum tour guide types, thus proving to be useful to museum management.

Based on the results, the study is beneficial to museum managers as they could improve their connection with audiences and tourists in general: the study provides evidence of the importance of tour guide performances, indicating that museum managers have to

pay more attention to museum tour guides and their performances and make a stronger connection between museum and visitors. Research is generally universal because all tourists, specifically those who travel abroad, need appropriate pampering and dedication during the visit, etc. In that sense, museums are at the beginning of new boom: one happened in the early 1990s, bringing virtual tours and lack of live interpretation into the museums. In our opinion, which was also partly proven with this study, the new era will come with a new live reinterpretation of museums: visitors need some live experience, which could include AR, VR, etc. Museum tour guides remain of paramount importance. Moreover, the study provides information important for the development of the training services of museum guides, directed towards the improvement of those performances that are of the most importance to visitors.

This study is significant on so many levels, having in mind a wide range of its applications; yet, it has a few limitations that can lead the research into some different areas. First, the research study applied an already confirmed scale designed for tour guides in general. Despite the fact that it has proven to be useful and capable of revealing certain tour guide performance dimensions, it also showed the necessity to develop and test specific scales aimed just for museum tour guides. Furthermore, it would be useful to conduct self-assessment of museum tour guides when their performance is taken into account. Finally, after validating a museum tour guides scale, it would be interesting to conduct cluster analysis or LPA to investigate museum tour guide profiles. Moreover, taking into account trends in tourist services related to microsegmentation and personalization of services, the authors assume that the features of the guide, arranged in five categories, could be adjusted based on the specific expectation of the guided group or individuals. This means that one tour guide might have different approach depending on the features and expectations of visitors, e.g., in guiding business groups or knowledge-oriented adults, the tour guide could expose more features of Classic Professional (CP). However, such an assumption needs some empirical evidence, which is why we consider it as an important direction of our future research.

**Author Contributions:** Conceptualization, Ž.A., S.K., M.D.V., U.S.; Data curation, Ž.A., M.D.V.; Formal analysis, S.K., U.S.; Investigation, S.K., M.B.; Methodology, Ž.A., M.B., S.K., M.D.V., U.S.; Resources, V.P.; Supervision, U.S., A.D., T.P., A.I.D., L.I.B., Z.Z.; Validation, Ž.A., M.B., M.D.V., A.I.D.; Writing—original draft, Ž.A., S.K.; Writing—review & editing, Ž.A., M.D.V., S.K., S.Đ.V.; A.D., L.I.B. All authors have read and agreed to the published version of the manuscript.

**Funding:** This research was supported by The Science Fund of the Republic of Serbia, Project No. 7739076, Tourism Destination Competitiveness—evaluation model for Serbia—TOURCOMSERBIA and by the Autonomous Province of Vojvodina, Provincial Secretariat for Higher Education and Scientific-Research Activity, Program 0201, project number: 142-451-2687/2021-01/01 and the project number 142-451-2615/2021-01/1.

**Institutional Review Board Statement:** Not applicable.

**Informed Consent Statement:** Not applicable.

**Conflicts of Interest:** The authors declare no conflict of interest.

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
