# Peer review of "Museum Tour Guide Performance: A Visitor Perspective"

_sustainability, doi:10.3390/su141610269_

Round 1

Reviewer 1 Report

The article presents the results of interesting research focused on the perception of various types of tourist guides by museum visitors. The text, however, requires some additions and clarifications:

 1) The description of the research method should be supplemented with the following information:
- characteristics of the respondents - did they visit the museum individually or with an organized group? Were there people with disabilities among the research participants?
- whether museums had the option of individual visits (e.g. with Audio Guide), i.e. without a guide or only with the guide?
- in what language were the questionnaires carried out with foreigners, did they fill in the questionnaire themselves, or did the researcher complete it?
2) Taking into account trends in tourist services related to microsegmentation or personalization of services, the question arises whether the features of the guide, arranged in five categories, are permanent features, or whether a given guide adjusts its approach depending on the features and expectations of visitors, e.g. in guiding business groups or knowledge-oriented adults exposes more features of Classic Professional (CP), and when showing fun-oriented holiday tourists it exposes more features of Agile Empath or Operational Erudite (OE)? 3). It is worth deepening the conclusions. Editorial notes:
- Tab. 3 - should be improved graphically
- repetitions appear, e.g. the ending takes the form of a summary rather than a substantive summary of the key, new knowledge.

Author Response

Please, find in attach response.

Authors 

Reviewer 2 Report

Thank you for submitting your study to Sustainability. This is certainly a niche area within tourist guiding research that longs for fresh insights – and your study addresses this omission in the literature.

The paper is clearly communicated and the empirical study seems to be well designed. However, its presentation in the manuscript does not really hang together. In particular, methodological section is weak and needs to be much better structured and more precise than what it currently offers. The conceptual framework is fine, however, I would like to see a more robust analysis of Huang et al’s work as the study heavily builds on it. Finally, what are the links between your study and sustainability? The submission of the manuscript to this journal needs to be justified through linking the research to some (or all) aspects of sustainability, and I would like to see the authors address this in revisions (both in the introduction and conclusion).

I note some comments below which I hope will help you improve the paper.

Introduction

Page 1 Line 2 check spelling scares -> scarce

Final para check spelling tightly-nit -> tightly-knit

In the first paragraph you mention ’early papers’ however you cite only one work. Can you add a couple more to justify this claim?

Tour (museum) guide performances

Para 1 – can you elaborate on what you suggest here? How is this relevant to the exploration of a museum tour experience?

Methods

The methods section needs to be more convincing as some information are currently hanging loose.

You state:

The study sample consists of the total of 255 visitors to cultural institutions – 123 domestic and 132 foreigners.

The authors of this paper hail from different countries, so it is unclear where do the domestic participants come from and who then you consider foreigners? It may become clear by looking at Table 2 that the study was conducted in Serbia as this country does not appear in the list, but this needs to be spelled out in text, and also in the abstract.

Also, in the same sentence, if you could be more precise when mentioning „cultural institutions“ because it is very a broad term. Could you mention museums instead?

Who conducted research and how did you gain access to museums to collect data etc, needs to be mentioned.

3. 1 Instrument

This needs to be further elaborated. Can you present Huang et al’s instrument and discuss how you built on/adapted it? Perhaps a visual presentation would be useful here.

3.2 procedure and analysis

Conducting an onsite questionnaire is fine. However you also mention interviews by which I was puzzled. Could you please explain what you mean by the following:

The questionnaire was distributed at the museums after the tour was completed. Each interview took approximately 10-12 minutes to complete.

Do you consider conducting a questionnaire an interview? You also mention survey. Please be more precise and consistent with terminology.

This is surprisingly large number of authors for this type of study, so mentioning the roles of co-authors in conducting the empirical study/analysis would be welcome.

Conclusions

Which research communities will benefit from your study? What is its sustainability element? What can you offer to the practice based on the findings?

Some general comments

Formatting needs to be adjusted (indent, the length of paragraphs etc).

I hope the comments are helpful and look forward to reviewing your revised manuscript.

Author Response

Please,

fin in attach response.

Authors

Reviewer 3 Report

Dear Authors:

1. Purpose and method are missing in the abstract.

2. In the article I miss a broader (international) presentation of the functioning and diversity of guide services. In some countries, a travel guide and a museum guide are completely different services. I have the impression that the authors are ignoring this fact.

3. The authors write that "Tour guiding is present in almost every tourist activity". Are there any studies that support this opinion? It is not clear from my observations.

4. The authors write "The study sample consists of the total of 255 visitors to" cultural institution ". It seemed to me that the article concerns museums. If so, it should be clarified.

5. The Sociodemographic characteristics of respondents range from 31-45 and 45-60. It should be 46-60.

6. Table 2 requires technical improvement. Also, I don't see the alphabetical order. Why is Croatia at the end?

7. Why is there no table that shows the income?

8. Table 3 requires technical improvement

9. The conclusions lack a broader reflection on whether the research is truly universal and can be used in other countries. Maybe such research is being conducted? I also lack information whether the results are applicable in practice.

11. This study was a one-time done. Will it be continued? If so, who would pay for it? Do you know how the museum will use the results?

12. I wonder if this article adds much to the theory. Can the problem of guide services not be linked to numerous studies on the quality and quality management of services?

Regards

The reviewer

Author Response

(The authors gave the same response as above.)

Round 2

Reviewer 2 Report

Thank you for revising the manuscript, it now reads much more convincing, better argued and linked to the journal's theme, and it now makes a contribution to the journal and tour guiding scholarship. I appreciate the way in which you responded to the reviewers' comments, and have no further concerns.

There are perhaps some minor stylistic issues (information rather than info) which the authors should attend to, but in general I am happy to recommend the manuscript for publication.

Reviewer 3 Report

I see the progress. Table 3 has technical mistakes.